# Development of the Intestinal Microbiota of Dairy Calves and Changes Associated with *Cryptosporidium* spp. Infection in Brazil

**DOI:** 10.3390/microorganisms12091744

**Published:** 2024-08-23

**Authors:** José Antônio Bessegatto, Júlio Augusto Naylor Lisbôa, Felippe Danyel Cardoso Martins, Roberta Lemos Freire, Elias Jorge Facury Filho, Amauri Alcindo Alfieri, Marcio C. Costa

**Affiliations:** 1Department of Clinical Sciences and Preventive Veterinary Medicine, Universidade Estadual de Londrina, Rodovia Celso Garcia Cid (PR 445) Km 380, Londrina 86057-970, PR, Brazil; bessegatto.ja@gmail.com (J.A.B.); janlisboa@uel.br (J.A.N.L.);; 2Department of Veterinary Medicine and Surgery, Universidade Federal de Minas Gerais—UFMG, Av. Antônio Carlos 6627, Belo Horizonte 31270-901, MG, Brazil; 3Department of Veterinary Biomedical Sciences, Université de Montréal, 3200 Sicotte, St-Hyacinthe, QC J2S 2M2, Canada

**Keywords:** calf diarrhea, microbiome, cryptosporidiosis

## Abstract

*Cryptosporidium* spp. is one of the most important pathogens infecting nursing calves worldwide. This study aimed to investigate the intestinal microbiota of dairy calves during the first month of life and the impact of diarrhea caused by *Cryptosporidium* on a Brazilian farm. Fecal samples from 30 calves were collected during the first month of life, and fecal scores were recorded. Samples from the second, third, and fourth days of life were analyzed by DNA sequencing of the 16S rRNA gene. In addition, samples of sixteen calves positive for *Cryptosporidium* spp. were retrospectively chosen according to the development of diarrhea: four and two days before diarrhea, at the onset of diarrhea, after four days of diarrhea, at the end of diarrhea, and after six days of diarrhea resolution. Diarrhea was observed in all calves (100%), starting at day 5 of life, and all calves tested positive for *Cryptosporidium* in at least one sample. The microbiota richness increased with age but was retarded by diarrhea. Compositional changes associated with *Cryptosporidium* infection included increases in *Fusobacterium, Prevotella*, and *Peptostreptococcus*, as well as decreases in *Collinsella* and Lachnospiraceae. In conclusion, *Cryptosporidium* infection has the potential to decrease richness and change the composition of the intestinal microbiota of dairy calves.

## 1. Introduction

Diarrhea is the most frequent condition observed in neonatal dairy calves worldwide, and various pathogens may elicit the illness, including bacteria, protozoa, and viruses [1]. The protozoa *Cryptosporidium* spp. has been identified as the major pathogen causing diarrhea in dairy calves, particularly in neonates, infecting almost 100% of calves in some regions [2]. In Brazil, the disease is present in up to 90% of farms, with between 11 and 63% of calves being infected within their first month of life [3,4].

Cryptosporidiosis poses significant challenges to the calf’s health and farm economics. The disease can significantly impact performance and increase the costs of veterinary care and medications, as clinical signs include severe diarrhea, dehydration, malnutrition, and increased mortality [5]. Moreover, *Cryptosporidium* is a zoonotic pathogen, posing a risk to human health through contaminated water and direct contact with infected animals [6].

An infection with few oocysts can induce parasite shedding and initiate diarrhea. The parasites’ infective form promotes self-infection and environmental contamination, where protection mechanisms enable the parasites to remain viable until ingested [7]. The protection conferred by colostrum against the parasite is weak and dependent on several factors, which makes it difficult to control the disease [8].

Recently, the role that the intestinal microbiota plays in gut health has received much attention. In calves, a gradual maturation of the microbiota is observed [9,10], but the presence of diarrhea can greatly impact its normal composition [11]. Age and diet, among many other factors, have repeatedly been shown to influence the composition of calves’ intestinal microbiota.

Calf diarrhea is usually a multifactorial condition, in which *Cryptosporidium* spp. infection is present concomitantly with other etiological agents, including viruses [12,13]. The dysbiosis found in animals with diarrhea has been associated with inflammation that can increase the length of clinical signs. Once the cycle is broken, by, for example, the administration of a fecal microbiota transplant (FMT), even viral diarrhea can quickly improve [14]. In fact, FMT has been reported to resolve diarrhea in calves successfully [15], but there are some conflicting results [16]. Those discrepancies might be associated with the adequate selection of donors and the presence of important metabolites in the transplanted solution [17]. Therefore, it is essential to have a good understanding of the microbiota changes associated with specific conditions (i.e., different etiologies) that could guide future interventional studies aiming to treat or prevent neonatal calf diarrhea.

DNA sequencing has also been used to characterize the microbiota of calves with experimentally induced [18] or naturally occurring cryptosporidiosis [19,20]. In goats, experimental infection demonstrated a decrease in the abundance of butyrate producers, which might further contribute to the reduction in body weight and perpetuation of diarrhea [21]. It has been suggested that the composition of the microbial profiles might influence the colonization by *Cryptosporidium* spp. [2]. Moreover, it has been suggested that bacterial markers present in the microbiota of young calves could predict susceptibility to *Cryptosporidium* spp. infection [22]. However, the few studies using new DNA sequencing technologies to investigate the important interaction between the gut microbiota and *Cryptosporidium* infection preclude major conclusions from being drawn. 

This study aimed to investigate the intestinal microbiota dynamics during the first days of life of dairy calves and the impact of diarrhea caused by *Cryptosporidium* spp.

## 2. Materials and Methods

All the experimental procedures were approved by the Ethics Committee in Animal Use of the Universidade Estadual de Londrina (protocol 3843.2017.60). The study was developed in a commercial dairy farm in Minas Gerais state, Brazil. The farm was selected based on its history of neonatal diarrhea caused by *Cryptosporidium* spp.

### 2.1. Animals

Thirty dairy heifers born on the farm were enrolled. The breed of the heifers was mainly Holstein, along with crossbred Brown Swiss–Jersey or Holstein–Gir.

All dams were vaccinated against rotavirus (serotypes 6 and 10) and *Escherichia coli* J5 (Rotatec J5; Biogénesis Bagó^®^, Buenos Aires, Argentina) in pre-established periods of the year (February, June, and October), independent of the dam’s gestational period. The dry-off period started 45 days before the parturition date. Dams were housed in a compost barn and without a strict calving area.

All calves received four liters of colostrum within six hours after birth. The colostrum was evaluated with a densitometer (colostrometer) and poor-quality colostrum was discarded. Free access to water and feed starters was provided.

Calves were raised in a collective rearing system on litter composed of sawdust with a layer of hay. Once a week, all hay was changed and sawdust was turned to allow the fermentation of organic matter. The neonates remained in collective pens with access to a solarium area until 25 days of life, after which they had access to a pasture area composed of *Cynodon dactylon*.

Diarrheic calves were offered oral electrolyte solutions according to their dehydration status and weight. Antibiotics were used only if calves were depressed, showed signals of septicemia, had a temperature higher than 40 °C, or were not responding to rehydration.

### 2.2. Sampling

Fecal samples were collected daily for 30 days directly from the rectum by digital stimulation after the morning meal.

All stools were scored from 0 to 4 at the moment of sampling (0—firm; 1—pasty, form maintained; 2—pasty, no defined form; 3—diarrhea, liquid stool with little solid components; 4—diarrhea, liquid stool without solid components). Samples with scores 3 and 4 were considered diarrheic. Feces were stored at −20 °C for up to 30 days and then transferred into a −80 °C freezer until analysis.

All samples were tested for the presence of *Cryptosporidium* spp. with PCR to investigate the parasite’s shedding.

Samples from the second day of life (*n* = 30) and samples from the third (*n* = 12) or fourth (*n* = 18) day of life were sequenced with 16S metabarcoding to characterize the transition of the postnatal microbiota of dairy calves and in an attempt to find early-life markers that could predict the occurrence of diarrhea and the susceptibility to *Cryptosporidium* spp.

In addition, to characterize the microbiota changes during diarrhea caused by *Cryptosporidium* spp. infection, samples from sixteen calves positive for *Cryptosporidium* were retrospectively chosen according to the development of diarrhea: four (D − 4) and two days before diarrhea (D − 2), at the onset of diarrhea (D0), after four days of diarrhea (D + 4), at the end of diarrhea (DX), and after six days of diarrhea resolution (DX + 6). For this analysis, samples were randomly selected from animals that had feces collected at all the proposed times. The selection of a subset of 15 calves was based on the availability of funding for the study.

### 2.3. DNA Extraction and Pathogen Detection

DNA extraction from fecal samples was performed with an EZNA stool kit (Omega, Bio-tek, Norcross, GA, USA) according to the manufacturer’s instructions.

The presence of *Cryptosporidium* spp. was determined by a PCR targeting an 18S rRNA region [23]. The primers for the first reaction were Fw1 (5′-TTCTAGAGCTAATACATGCG-3′) and Rv2 (5′-CCCATTTCCTTCGAAACAGGA-3′). The second reaction was performed using the primers Fw3 (5′-GGAAGGGTTGTATTTATTAGATAAAG-3′) and Rv4 (5′-AAGGAGTAAGGAACAACCTCCA-3′). Reactions consisted of 1 × PCR buffer, 2.5 mM of MgCl2, 0.2 µM of DNTP, 0.2 µM of each primer, 1U of Platinum Taq (Invitrogen, Waltham, MA, USA), 2 µl of DNA, and sterilized ultrapure water (total volume of 25 µL). The same cycling conditions were used in both reactions: 95 °C for 5 min, followed by 35 cycles of 94 °C for 45 s, 55 °C for 45 s, and 72 °C for 1 min, after a final extension of 72 °C for 5 min. *Cryptosporidium parvum* DNA and ultrapure water were included as positive and negative controls in all batches, respectively. Products from the second amplification were submitted to electrophoresis in a 1.5% agarose gel stained with SYBR^®^ Safe (Invitrogen) and photo-documented by the LPix Image ST Software v.2.11 (Loccus Biotechnology, Cotia, Brazil).

To verify infections by other agents in the diarrhea, samples from D0 were submitted for testing for Bovine Coronavirus, Bovine rotavirus A, *Salmonella enterica*, and *Escherichia coli* K99. The presence of Bovine rotavirus A was assessed by the polyacrylamide gel electrophoresis (PAGE) technique and silver staining [24]. The diagnosis of Bovine Coronavirus was made using a semi-nested PCR by the amplification of the N gene according to Takiuchi et al. [25]. To detect Salmonella, a pair of primers from the *inv*A gene were used [26], and the PCR was carried out according to Malorny et al. [27]. For the presence of *Escherichia coli* K99, PCR was carried out as previously described [28,29].

For the microbiota analysis, the V4 region of the 16S rRNA gene was amplified by a two-step PCR [30]. Sequencing was performed using an Illumina MiSeq IEMFile version 4 platform, using a reagent kit V2 (2 × 250 cycles) (San Diego, CA, USA) at the Genome Quebec Innovation Centre (Montréal, QC, Canada).

The software Mothur (version 1.46.1) was used for bioinformatic analysis following a standard protocol [31]. Good quality reads with more than 97% similarity were clustered into operational taxonomic units (OTUs) and classified according to the Ribosomal Database Project (RDP) databank. Reads classified in the same genus were further merged into phylotypes for downstream analysis.

Alpha diversity was characterized by the number of observed genera, the Chao richness estimator, and the Shannon and Inverse Simpson indices. Beta diversity, which is the comparison between the microbiota composition present in each sample, was evaluated using the Jaccard index as a measure of the membership and the Yue and Clayton index for community structure. The first considers only the presence or absence of each taxon in the community, and the second also considers the abundance of each population. The Principal Coordinate Analysis (PCoA) was utilized to visualize the similarities between communities’ membership and structure.

### 2.4. Statistical Analysis

Alpha-diversity indices and the relative abundance of the most abundant phyla and genera were compared using a One-Way ANOVA with the Bonferroni test as a multiple comparison correction to evaluate the differences between moments of diarrhea (D − 4, D − 2, D0, D + 4, DX, and DX + 6) and between days of life (Day 2, Day 3, and Day 4). Statistical significance was considered for *p* values < 0.05 unless otherwise recommended by the Bonferroni test. A *t*-test was used to compare the taxa of samples that were positive or negative for *Cryptosporidium* spp. at 3 and 4 days of life.

The analysis of molecular variance (AMOVA) was used to verify whether there was a difference in beta diversity between groups, and the Bonferroni test was used to account for multiple comparison errors.

## 3. Results

### 3.1. Incidence of Diseases

Diarrhea was observed in all calves (100%) during the first month of life. The main episode of diarrhea was observed to start on day 5 and ending on day 16 (Figure 1), lasting an average of 7.86 days (min: 3 days; max: 12 days). The percentage of calves positive for *Cryptosporidium* spp. is also presented in Figure 1. Virtually all of the samples collected between days 5 and 14 contained *Cryptosporidium* spp. The testing for Bovine Coronavirus, Bovine rotavirus A, *Salmonella enterica*, and *E. coli* K99 yielded negative results for all samples.

Nine calves with diarrhea were treated with Sulfadoxine/Trimetropina (20 mg/kg) and five with enrofloxacin (10 mg/kg) for three days. Eight calves (26.67%) developed pneumonia during their first month of life and were treated with enrofloxacin (12.5 mg/kg; once) or penicillin (40,000 UI/kg; three days).

### 3.2. Microbiota Analysis

Sequencing of the V4 region of the 16S rRNA gene produced 5,440,437 good-quality reads from 125 samples. A subsample of 19,064 reads per sample was used for normalization, yielding an average of 99.95% of the Good’s coverage index.

The initial analysis included samples from 30 calves collected on the second day of life (Day 2) and from 12 and 18 calves from the third (Day 3) and fourth (Day 4) days of life, respectively. Alpha-diversity was evaluated by the number of genera and the Chao, Inverse Simpson, and Shannon indices. All indices significantly increased from Day 2 to Day 4 (Figure 2A).

The presence of Cryptosporidium spp. in samples collected on Day 3 (Positive = 3; Negative = 9) was associated with greater abundances of Peptostreptococcus (19.10% vs. 2.51%; *p* = 0.018) and at Day 4 (Positive = 9; Negative = 9) with Bacteroidetes (27.50% vs. 11.50%; *p* = 0.038) and Prevotella (13.10 vs. 4.21%; *p* = 0.045), but with lower abundances of Actinobacteria (1.61 vs. 7.95; *p* = 0.022), Lachnospiraceae (2.75% vs. 13.40%; *p* = 0.008), and Collinsella (1.54% vs. 7.80%; *p* = 0.021).

For the diarrhea analysis, samples from D − 4 coincided with samples collected between 2 and 4 days of life; samples from D − 2 varied between 3 and 5 days of life; samples from D0 varied between 6 and 7 days of life; samples from D + 4 varied between 9 to 11 days of life; samples from DX varied from 12 to 16 days of life; and samples from DX + 6 varied from 18 to 23 days of life. The analysis of samples collected before, during, and after the occurrence of diarrhea is presented in Figure 2B. Upon the onset of diarrhea (D0), calves had significantly higher richness than four days before. That was followed by a non-significant decrease after four days of diarrhea and a subsequent significant increase after the resolution of the condition. Although not statistically significant, the same trend was observed in the Simpson index.

The results from the beta diversity analysis (AMOVA), which compares microbiota composition, showed marked differences in community membership (which considers only the presence or absence of each bacterium) between calves at 2, 3, and 4 days of life (all *p* < 0.001). For community structure (which also considers the abundance of each bacterium), statistical significance was observed between D2 and D4 (*p* < 0.001) and D3 and D4 (*p* = 0.007), but not between D2 and D3 (*p* = 0.048; Bonferroni correction = 0.016).

Firmicutes, Proteobacteria, Fusobacteria, Bacteroidetes, and Actinobacteria were the most abundant phyla on Days 2, 3 and 4. Significant increases from Day 2 to Day 4 were observed for the phyla Firmicutes, Bacteroidetes, Actinobacteria, and Verrucomicrobia, to the detriment of the decrease in Proteobacteria (Appendix A). The most abundant genera observed were Escherichia/Shigella, Fusobacterium, unclassified Enterobacteriaceae, Lactobacillus, and Bacteroides. Significant increases from Day 2 to Day 4 were observed in unclassified Lachnospiraceae, Prevotella, Faecalibacterium, Phascolarctobacterium, Collinsella, Blautia, and Alloprevotella. The genera that significantly decreased were Escherichia/Shigella, unclassified Enterobacteriaceae, and Clostridium sensu stricto (Appendix A).

Considering the progression of diarrhea, there were also marked differences in samples collected before, during, and after diarrhea resolution for both community membership and structure. Figure 3 presents the Principal Coordinate Analysis (PCoA) representing the similarity among the bacterial microbiota of calves at different moments of diarrhea evolution. Statistical significance was observed in all comparisons of membership (all *p* < 0.001, but the comparison of D0 and D − 2: *p* = 0.003). For community structure, there were significant differences between D − 2 and D2, D − 2 and D4, D0 and D2, D0 and DX + 6, D2 and D4, D2 and DX, D2 and DX + 6, and D4 and DX + 6 (all *p* < 0.001).

Regarding samples collected before, during, and after diarrhea, there was a significant increase in Firmicutes and Bacteroidetes from D − 4 to DX + 6. Actinobacteria increased from D − 4 to D − 2, while Proteobacteria decreased from D − 4 onwards. In addition, a high abundance of Fusobacteria was associated with the presence of diarrhea (D + 4) but decreased after diarrhea ceased (DX and DX + 6) (Figure 4).

In concordance with the phylum level analysis, statistical differences at the genus level included Escherichia/Shigella and an unclassified Enterobacteriaceae that were increased on D − 4; Prevotella, Collinsella, Phascolarctobacterium, and Alloprevotella that were decreased on D − 4; Blautia that was increased after DX; Peptostreptococcus and an unclassified Lachnospiraceae that were increased in D + 4; Fusobacterium that were decreased on DX + 6; and Clostridium sensu stricto and Streptococcus that were decreased on DX + 6.

## 4. Discussion

The present study used 195 fecal samples to evaluate the dynamics of neonatal calf microbiota and its associations with *Cryptosporidium* spp. infection. The marked increase in bacterial richness and diversity found in newborns during their first days of life supports the findings of other studies [32,33]. The results from the beta diversity analysis, which compares microbiota composition, showed marked differences in membership and structure between calves at 2, 3, and 4 days of life. Taken together with the results of the alpha diversity analysis, the present study demonstrated a dynamic microbial shift in the early life of dairy calves. This shift included a decrease in Proteobacteria, which was highly abundant at 2 days of life. Members of this phylum, including *Escherichia* spp., might be acquired during birth [34] and are oxygen consumers responsible for providing an anaerobic environment [35] to the intestines.

The present study showed that *Cryptosporidium* spp. virtually infected all animals between 5 and 14 days of life, and more than 50% of calves remained positive until the last sampling, when they were 30 days old, even though diarrhea had resolved. This finding highlights the persistence of the parasite and the importance of environmental contamination. It has been shown that a higher intake of oocysts is associated with earlier onset of diarrhea [36] and with the incidence of watery diarrhea [37,38].

*Cryptosporidium parvum* is one of the most common pathogens infecting and causing diarrhea worldwide in dairy calves under one month of age [39,40]. The prevalence of infection can be highly variable between herds but can reach up to 100% of animals, as observed in our study. The incidence of diarrhea observed in the studied calves also strongly agreed with the reported age of susceptibility and recovery [41].

The results of the microbiota analysis showed an increasing richness over time, which is consistent with the current literature [20,42,43]. Importantly, it could be observed that the increasing trend in richness and diversity (Simpson) was delayed by a dissonant lower (albeit not statistically significant) richness and diversity during the presence of diarrhea (Figure 2). Furthermore, there was a statistical increase in those indices 6 days after the resolution of diarrhea, supporting that diarrhea causes a delay in the physiological maturation of the fecal microbiota of calves. Lower intestinal bacterial diversity during *Cryptosporidium* infection has been previously reported in calves and goat kids [20,21].

This study also sought to evaluate potential changes in the microbiota before the onset of diarrhea (D − 4) that could predict the occurrence or severity of the condition. However, as most of the calves developed diarrhea at 7 days of age, D − 4 coincided with samples collected at 2 days of age, a moment in which the composition of the microbiota is actively changing due to the physiological maturation of the gut and appearance of a strictly anaerobic environment. In fact, there were many significant changes at the phylum level between samples collected at D − 4 and diarrhea recovery, but the incidence of 100% of calves with diarrhea at days 5–6 of life makes it impossible to determine if those differences were physiological or actually caused by diarrhea.

It has been suggested that early changes in the microbiota of dairy calves could predict the susceptibility to *Cryptosporidium* spp. infection [22]. However, this hypothesis could not be tested in the present study because all calves tested positive for the presence of the parasite. Nevertheless, the comparison of samples collected at 3 and 4 days of life revealed associations between certain bacteria taxa and the presence of *Cryptosporidium* infection, such as greater abundances of *Prevotella* and *Peptostreptococcus* and lower abundances of *Collinsella* and Lachnospiraceae. Interestingly, *Peptostreptococcus* and *Collinsella* have been associated with inflammation and lower weight gain in calves infected with *Cryptosporidium* [20]. Other studies reported differences in the microbiota of mice experimentally infected with *Cryptosporidium parvum* at five days of life without developing diarrhea [44,45]. However, studies performed in laboratory animals reared in a controlled environment might not be ideal for comparison with the progression of the disease in calves. 

Interestingly, among the bacteria statistically associated with diarrhea in the present study, *Fusobacterium* has also been reported to be highly abundant in calves with cryptosporidiosis [19,20]. Ichikawa-Seki et al. (2019) showed a positive correlation between *Cryptosporidium parvum* oocysts count and watery diarrhea with increased *Fusobacterium* in calves [19]. Furthermore, Dorbek-Kolin et al. (2022) also found positive associations between oocyst count and the abundance of *Fusobacterium* [20]. The role of *Fusobacterium* in the pathogenesis of calf cryptosporidiosis remains to be determined. Still, it might be related to epithelial damage caused by *Cryptosporidium* with subsequent mucosal-associated dysbiosis, opportunistically favoring the increase in *Fusobacterium* spp. Moreover, it is possible that the dysbiosis observed during diarrhea is a consequence of local inflammation, as several inflammatory markers have been reported to be increased during the acute phase of the infection, including a positive correlation of serum amyloid A with the abundance of *Fusobacterium* [20]. Nevertheless, the correction of dysbiosis has been associated with faster recovery of viral diarrhea in puppies, suggesting that dysbiosis itself might play a role in the perpetuation of diarrhea [14]. Therapies with microbiota transplantation are in their infancy in calves but might become interesting strategies in the future [15,17,46], but the results in the literature remain in conflict with each other [16].

The present study brings novel information about the microbiota of young calves with diarrhea caused by *Cryptosporidium*, but some limitations must be acknowledged. First, the DNA sequencing methods used are not quantitative; therefore, they do not allow inferences about the real number of bacteria. For instance, it is possible that the increase in the relative abundance of *Fusobacterium* was caused by a decrease in the other populations rather than a real absolute increase in the bacterium. Another limitation was the use of antibiotics in some calves. Although a post hoc analysis revealed no difference in the microbiota composition of treated versus untreated animals, this could be a source of data variability. The use of a homogeneous population not exposed to many variables is essential for microbiota studies. As such, the present study was performed on one single farm to avoid the introduction of another variable. Therefore, larger studies are necessary before the results can be extrapolated. Nevertheless, our results are consistent with those of other studies in the literature.

## 5. Conclusions

We concluded that *Cryptosporidium* spp. is an important diarrhea pathogen in dairy calves, and infection alters the intestinal microbiota. Those changes included potentially decreased richness and diversity, increases in *Fusobacterium*, *Prevotella*, and *Peptostreptococcus*, and decreases in *Collinsella* and Lachnospiraceae. Further studies evaluating approaches aiming to modulate the intestinal microbiota (e.g., probiotics, prebiotics, postbiotics, etc.) are justified.

## Figures and Tables

**Figure 1 microorganisms-12-01744-f001:**
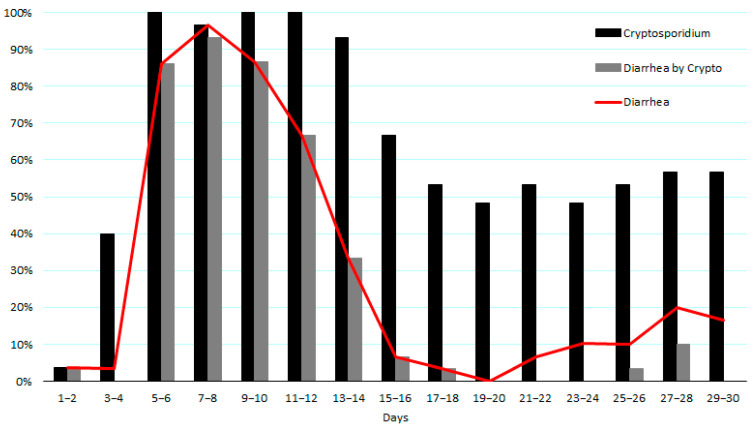
Incidence of diarrhea during the first month of life of 30 dairy calves and the percentage of samples positive for *Cryptosporidium* spp., as well as the incidence of diarrhea attributed to Cryptosporidium infection.

**Figure 2 microorganisms-12-01744-f002:**
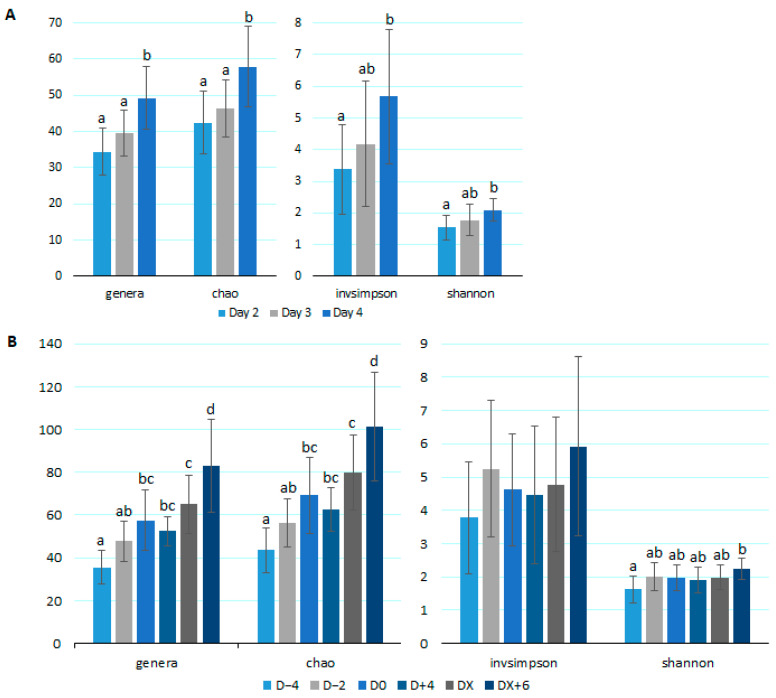
Results of alpha diversity analysis indicated by the number of genera, Chao, Inverse Simpson (invsimpson), and Shannon indices found in fecal samples from dairy calves at 2, 3, and 4 days of life (**A**) and 4 days (D − 4) and 2 days (D − 2) before the development of diarrhea (D0), 4 days after (D + 4), on the day of resolution (DX), and 6 days later (DX + 6) (**B**). Different letters between samplings indicate statistical differences (*p* < 0.05).

**Figure 3 microorganisms-12-01744-f003:**
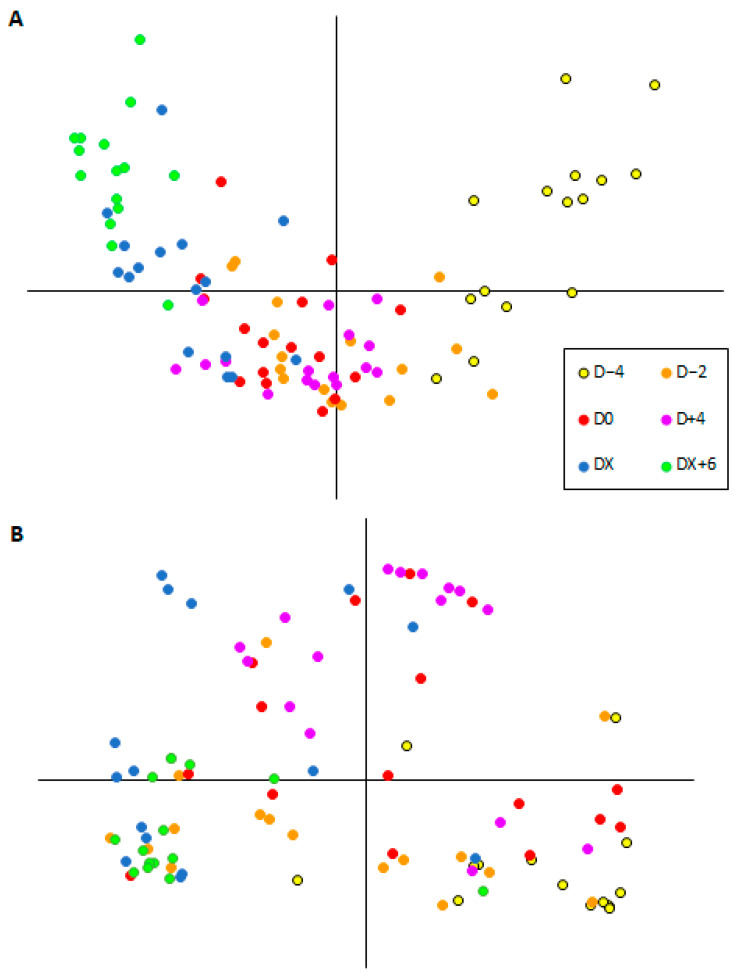
Principal Coordinate Analysis plot representing microbial communities of fecal samples from dairy calves in different moments of diarrhea episodes. (**A**) Membership (Jaccard). (**B**) Structure (Yue and Clayton). D − 4: four days before the development of diarrhea; D − 2: two days before diarrhea; D0: onset of diarrhea; D + 4: four days after the onset of diarrhea; DX: resolution of diarrhea; and DX + 6: six days after resolution.

**Figure 4 microorganisms-12-01744-f004:**
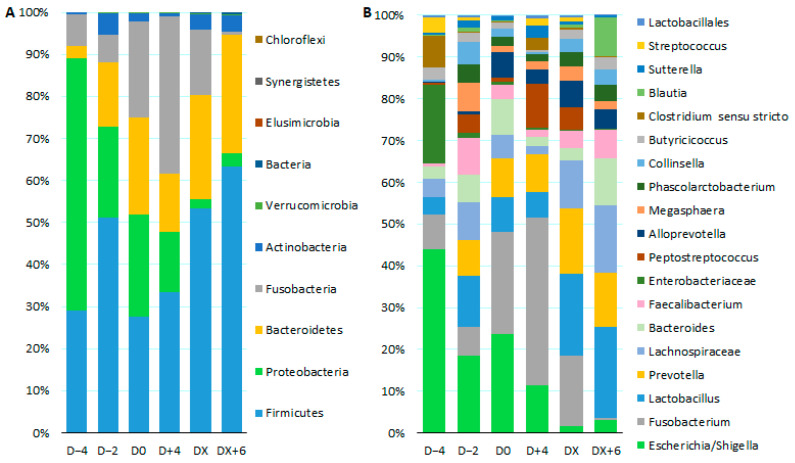
Relative abundance of the main phyla (**A**) and genera (**B**) found in fecal samples from dairy calves collected at different moments of diarrhea episodes. D − 4: four days before the development of diarrhea; D − 2: two days before diarrhea; D0: onset of diarrhea; D + 4: four days after the onset of diarrhea; DX: resolution of diarrhea; and DX + 6: six days after resolution of diarrhea.

## Data Availability

The original contributions presented in the study are included in the article/Appendix A, further inquiries can be directed to the corresponding author.

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
