# Peer review of "Development of the Intestinal Microbiota of Dairy Calves and Changes Associated with Cryptosporidium spp. Infection in Brazil"

_microorganisms, 2024, doi:10.3390/microorganisms12091744_

Round 1

Reviewer 1 Report

Comments and Suggestions for Authors

microorganisms-3062666

Development of the intestinal microbiota of dairy calves and changes associated with Cryptosporidium spp. infection.

José Antônio Bessegatto et al.

The authors provide convincing evidence that the diarrhea observed in the study calves is likely caused by Cryptosporidium infection and then proceed to present a timeline study of microbiome changes pre and post illness. Overall, I found the study was well designed and the paper well written and easy to follow.

In my view the only weakness was a lack of controls i.e. calves that did not get a Cryptosporidium infection (or other diarrhea) in the experiment and therefore we do not know how the ‘normal’ calf microbiome develops undisturbed. It would have been nice to have this comparison, but I think the authors have considered this in their discussion and addressed this issue well. Diarrhea does coincide with reduced alpha diversity, but as the author point out, this may also be due to naturally occurring changes i.e. the physiological maturation of the gut and appearance of a strictly anaerobic environment (line 290) after D-4. This weakens the conclusion that microbiota richness was delayed directly by diarrhea / Cryptosporidium infection.

Line 157: It says AMOVA was used but there are no AMOVA results presented that I could find. I am not sure how AMOVA applies to this study? It is typically used for population genetics.

Minor suggestions:

Line 23 here and throughout and italicize Cryptosporidium and other genus names.

Line 41: suggest removing the word “already”

Line 74: “All calves received four liters of colostrum until six hours after birth” I found the wording of this sentence confusing, suggest rephrasing for clarity. 

Line 87: delete )

Line 93: suggest adding “with PCR” after “Cryptosporidium spp.”

Line 96: suggest adding “with 16S metabarcoding” after “were sequenced”.

Line 141: Correct Alpha typo

Line 224: suggest changing evolution to progression.

Author Response

We would like to deeply thank the reviewers for their time in revising our manuscript and for their comments, which were essential to elevating the standards of our study. We were glad to be able to address most of their comments and suggestions. Please see below our answers to each of their comments.

Reviewer 1:

The authors provide convincing evidence that the diarrhea observed in the study calves is likely caused by Cryptosporidium infection and then proceed to present a timeline study of microbiome changes pre and post illness. Overall, I found the study was well designed and the paper well written and easy to follow.

AUTHORS: Thank you for your positive feedback.

In my view the only weakness was a lack of controls i.e. calves that did not get a Cryptosporidium infection (or other diarrhea) in the experiment and therefore we do not know how the ‘normal’ calf microbiome develops undisturbed. It would have been nice to have this comparison, but I think the authors have considered this in their discussion and addressed this issue well. Diarrhea does coincide with reduced alpha diversity, but as the author point out, this may also be due to naturally occurring changes i.e. the physiological maturation of the gut and appearance of a strictly anaerobic environment (line 290) after D-4. This weakens the conclusion that microbiota richness was delayed directly by diarrhea / Cryptosporidium infection.

AUTHORS:  In fact, the samples were collected every other day in the hope that some of the calves could be crypto-negative and, therefore, used as controls. However, they all developed diarrhea at the same age and were positive for the parasite. Nevertheless, we compared our results with the ones in the literature. Regarding the decreased diversity, I have just reviewed two other manuscripts, one evaluating the microbiota of calves during crypto infection and the other just diarrhea without an established diagnosis, and both support our conclusions.

Line 157: It says AMOVA was used but there are no AMOVA results presented that I could find. I am not sure how AMOVA applies to this study? It is typically used for population genetics.

AUTHORS: All the p values presented for beta diversity analysis (Lines 217-222) result from the AMOVA test. This has been added to the text.

Minor suggestions:

Line 23 here and throughout and italicize Cryptosporidium and other genus names.

AUTHORS: The text has been modified.

Line 41: suggest removing the word “already”

AUTHORS: The text has been modified.

Line 74: “All calves received four liters of colostrum until six hours after birth” I found the wording of this sentence confusing, suggest rephrasing for clarity.

AUTHORS: The text has been modified for clarity “ All calves received four liters of colostrum within six hours after birth”

Line 87: delete )

AUTHORS: Thank you for noticing that.

Line 93: suggest adding “with PCR” after “Cryptosporidium spp.”

AUTHORS: Done.

Line 96: suggest adding “with 16S metabarcoding” after “were sequenced”.

AUTHORS: Done.

Line 141: Correct Alpha typo

AUTHORS: Thank you for noticing that.

Line 224: suggest changing evolution to progression.

AUTHORS: Done.

Reviewer 2 Report

Comments and Suggestions for Authors

The study is interesting to the extent that it complements the existing data in the specialized literature.

I think you should answer the question : if the treatments performed did not influence the microbiota?

The conclusions could be a little more developed.

The references could be completed with:

Ras et al. (2015) -Perturbation of the intestinal microbiota.....Int. J. Parasitol. and other, see Pub Med.

Author Response

We would like to deeply thank the reviewers for their time in revising our manuscript and for their comments, which were essential to elevating the standards of our study. We were glad to be able to address most of their comments and suggestions. Please see below our answers to each of their comments.

Reviewer 2:

The study is interesting to the extent that it complements the existing data in the specialized literature. I think you should answer the question: if the treatments performed did not influence the microbiota?

AUTHORS: Thank you very much for your comments. I assume the reviewer refers to the treatment with antimicrobials. That was, in fact, one of our concerns, and for this reason, we performed a post hoc analysis that revealed no impact of the antibiotics used in the calves’ microbiota. This has been reported in other studies, and it is possibly because the extent of dysbiosis caused by diarrhea is greater than the impact of antibiotics. This was stated in the discussion: ” Another limitation was the use of antibiotics in some calves. Although a post hoc analysis revealed no difference in microbiota composition of treated versus untreated animals, this could be a source of data variability.”

The conclusions could be a little more developed.

AUTHORS: We tried to keep the conclusions as sound as possible to avoid overstatements that are not supported by our results and, therefore, could mislead readers. In fact, this is very common in microbiota studies and we try to avoid that.  We have added the following sentence to the conclusions: “Another limitation was the use of antibiotics in some calves. Although a post hoc analysis revealed no difference in microbiota composition of treated versus untreated animals, this could be a source of data variability.”

The references could be completed with:

Ras et al. (2015) -Perturbation of the intestinal microbiota.....Int. J. Parasitol. and other, see Pub Med.

AUTHORS: Thank you for your suggestion. This study has been added to the discussion.

Reviewer 3 Report

Comments and Suggestions for Authors

The work is interesting and aims to see the relationship between Cryptosporidium and the intestinal microbiota in dairy calves.

There are some questions that need to be thought about and, if possible, added to the article to eliminate some problems that impact the results:

1. Antibiotic treatment of some newborns can alter the intestinal microbiota. These animals should be removed from the experiment.

2. the total number of animals studied is small when we exclude those that received antibiotics. In addition to the lack of controls since everyone was parasitized. Could you increase the sample size?

In figure 2 it is not clear who is statistically different from whom because the caption lacks an explanation of the letters above the bar.

The supplementary table should be in the body of the article and not separately.

Author Response

We would like to deeply thank the reviewers for their time in revising our manuscript and for their comments, which were essential to elevating the standards of our study. We were glad to be able to address most of their comments and suggestions. Please see below our answers to each of their comments.

Reviewer 3

The work is interesting and aims to see the relationship between Cryptosporidium and the intestinal microbiota in dairy calves.

There are some questions that need to be thought about and, if possible, added to the article to eliminate some problems that impact the results:

  1. Antibiotic treatment of some newborns can alter the intestinal microbiota. These animals should be removed from the experiment.
  2. the total number of animals studied is small when we exclude those that received antibiotics. In addition to the lack of controls since everyone was parasitized. Could you increase the sample size?

AUTHORS: We agree, as stated above in the answers to reviewer 2 and in the discussion of the manuscript, we tested to see if the microbiota of calves receiving antibiotics would have an outlier pattern, which was not the case. For this reason, we decided to maintain the samples in the analysis. Nevertheless, this is clearly stated as one of the study’s limitations. We don’t have samples collected from other calves that could allow us to increase the sample size. Nevertheless, despite the limitations of the study, our results are consistent with other reports and can contribute with the current knowledge in the field.

In figure 2 it is not clear who is statistically different from whom because the caption lacks an explanation of the letters above the bar.

AUTHORS: The legend of the figure states: “Different letters between samplings indicate statistical differences (p<0.05).” It means that if there is an “a” in one column and an “ab” in the other, they are not statistically different. They would differ from a column containing a “c,” for example. This is standard in our publications.

The supplementary table should be in the body of the article and not separately.

AUTHORS: Thank you for your remark. I have uploaded the table to the designated field during the submission process. I will let the editorial staff take care of this.

Reviewer 4 Report

Comments and Suggestions for Authors

The authors should put the location in both the title and the abstract.

The introduction is very short and unattractive to the reader. It doesn't draw any attention to a subject that has been extensively studied. What is the incidence and cases reported in the locality?

You only used one farm and with a number of 30. This is too small to support your conclusions.

Despite having worked with a modern technique, the methodology should have more diversity of environments and locations.

The titles of the figures should be self-explanatory.

The authors write in the discussion:

The present study brings novel information about the microbiota of young calves  with diarrhea caused by Cryptosporidium, but some limitations must be acknowledged.  Firstly, the DNA sequencing methods used are not quantitative; therefore, they do not  allow inferences about the real number of bacteria. For instance, it is possible that the increase in relative abundance of Fusobacterium was caused by a decrease in the other populations rather than a real absolute increase of the bacterium. Another limitation was the  Microorganisms 10 of 12 use of antibiotics in some calves. Although a post hoc analysis revealed no difference in  microbiota composition of treated versus untreated animals, this could be a source of data  variability. Nevertheless, our results are consistent with those of others in the literature.

frankly, this discussion makes the work, which is little discussed because of its limitations, unfeasible.

Author Response

We would like to deeply thank the reviewers for their time in revising our manuscript and for their comments, which were essential to elevating the standards of our study. We were glad to be able to address most of their comments and suggestions. Please see below our answers to each of their comments.

Reviewer 4

The authors should put the location in both the title and the abstract.

AUTHORS: This information has been added to the title and abstract.

The introduction is very short and unattractive to the reader. It doesn't draw any attention to a subject that has been extensively studied. What is the incidence and cases reported in the locality?

AUTHORS: I respect your opinion, which is different from ours and from the other three reviewers who found the manuscript to be well written. Nevertheless, we have expanded the introduction and added the requested information.

You only used one farm and with a number of 30. This is too small to support your conclusions.

AUTHORS: The assessment of the sample size should be based on the statistical analyses of the study rather than subjectively. The scientific community accepts numerous studies published in Nature using 6 animals per group based on their statistical analysis. Considering the total number of samples collected during the study, this is one of the largest to date investigating the longitudinal changes of the calf's microbiota. The changes observed over time are very clear, as there is little data variability within each sampling time. For those reasons, we believe our design is adequate and the conclusions supported by the results.

Despite having worked with a modern technique, the methodology should have more diversity of environments and locations.

AUTHORS: We agree with the reviewer that our data cannot be extrapolated to other geographical locations and other environments. However, the introduction of more farms would have to be treated as another variable, demanding an even greater number of animals. We have clearly stated that in the discussion as a limitation as follows: “The use of a homogeneous population not exposed to many variables is essential for microbiota studies. As such, the present study was performed on one single farm to avoid the introduction of another variable. Therefore, larger studies are necessary before the results can be extrapolated.”

The titles of the figures should be self-explanatory.

AUTHORS: The titles of the figures were revised, and we believe they are complete and self-explanatory. If the reviewer has any specific suggestion we will be happy to address it.

The authors write in the discussion:

The present study brings novel information about the microbiota of young calves  with diarrhea caused by Cryptosporidium, but some limitations must be acknowledged.  Firstly, the DNA sequencing methods used are not quantitative; therefore, they do not  allow inferences about the real number of bacteria. For instance, it is possible that the increase in relative abundance of Fusobacterium was caused by a decrease in the other populations rather than a real absolute increase of the bacterium. Another limitation was the  Microorganisms 10 of 12 use of antibiotics in some calves. Although a post hoc analysis revealed no difference in  microbiota composition of treated versus untreated animals, this could be a source of data  variability. Nevertheless, our results are consistent with those of others in the literature.

frankly, this discussion makes the work, which is little discussed because of its limitations, unfeasible.

AUTHORS: We respectfully disagree with the reviewer. One hundred percent of the studies evaluating the calf microbiome used the technique. So, if our statement makes the study unfeasible, so are all the others already published. As an expert in the field, I can ensure that short-read DNA sequencing has many limitations that are rarely mentioned in scientific articles. My philosophy is to address and explain those limitations to educate the scientific community, avoiding that imprecise conclusions are withdrawn from those studies.

Round 2

Reviewer 3 Report

Comments and Suggestions for Authors

The authors were unable to change the methodology by adding more animals and animals without antibiotic use. However, they explained this in their responses and also emphasized the issue of antibiotic use in the discussion and conclusion, placing this medication as a limitation of the study. Therefore, I think the changes made were good and the work can be published.

Reviewer 4 Report

Comments and Suggestions for Authors

Despite the modifications and justifications made by the authors, I insist, the article is weak. It has an insignificant n, it only worked on one farm, the methods don't support the conclusions and results, the article is poorly discussed and speculative. My final decision is to reject the paper. If three other reviewers have not done so and have opted to continue, it will not change my opinion, which is consistent.